# A First-Principles Study on the Multilayer Graphene Nanosheets Anode Performance for Boron-Ion Battery

**DOI:** 10.3390/nano12081280

**Published:** 2022-04-09

**Authors:** Mustapha Umar, Chidera C. Nnadiekwe, Muhammad Haroon, Ismail Abdulazeez, Khalid Alhooshani, Abdulaziz A. Al-Saadi, Qing Peng

**Affiliations:** 1Chemistry Department, King Fahd University of Petroleum and Minerals, Dhahran 31261, Saudi Arabia; g201708770@kfupm.edu.sa (M.U.); g201707330@kfupm.edu.sa (C.C.N.); muhammad.hanif@kfupm.edu.sa (M.H.); hooshani@kfupm.edu.sa (K.A.); asaadi@kfupm.edu.sa (A.A.A.-S.); 2Interdisciplinary Research Center for Membranes and Water Security, King Fahd University of Petroleum and Minerals, Dhahran 31261, Saudi Arabia; 3Interdisciplinary Research Center for Refining and Advanced Chemicals, King Fahd University of Petroleum and Minerals, Dhahran 31261, Saudi Arabia; 4Interdisciplinary Research Center for Hydrogen and Energy Storage, King Fahd University of Petroleum and Minerals, Dhahran 31261, Saudi Arabia; 5Physics Department, King Fahd University of Petroleum and Minerals, Dhahran 31261, Saudi Arabia; 6KACARE Energy Research and Innovation Center at Dhahran, Dhahran 31261, Saudi Arabia

**Keywords:** DFT, graphene layers, boron-ion battery, adsorption, reduced density gradient

## Abstract

Advanced battery materials are urgently desirable to meet the rapidly growing demand for portable electronics and power. The development of a high-energy-density anode is essential for the practical application of B^3+^ batteries as an alternative to Li-ion batteries. Herein, we have investigated the performance of B^3+^ on monolayer (MG), bilayer (BG), trilayer (TG), and tetralayer (TTG) graphene sheets using first-principles calculations. The findings reveal significant stabilization of the HOMO and the LUMO frontier orbitals of the graphene sheets upon adsorption of B^3+^ by shifting the energies from −5.085 and −2.242 eV in MG to −20.08 and −19.84 eV in 2B^3+^@TTG. Similarly, increasing the layers to tetralayer graphitic carbon B^3+^@TTG_asym and B^3+^@TTG_sym produced the most favorable and deeper van der Waals interactions. The cell voltages obtained were considerably enhanced, and B^3+^/B@TTG showed the highest cell voltage of 16.5 V. Our results suggest a novel avenue to engineer graphene anode performance by increasing the number of graphene layers.

## 1. Introduction

The past two decades have witnessed impressive improvements in lithium-ion battery (LIB) technologies for portable consumer electronics, electrical devices, and energy storage [1,2]. LIBs have demonstrated outstanding performance, including superior energy density, operating voltage, life cycle, and minimal rate of self-discharge, as well as low volume [3]. Generally, LIBs have exhibited exceptional performance over other known rechargeable ion battery systems [4,5,6]. More significantly, the 2019 Nobel Prize award in Chemistry was received by researchers in the field of lithium-ion batteries [7,8]. Despite the appealing applicability of LIBs, limitations such as narrow lifetime, inadequate performance at low temperatures, and most severely, the swift exhaustion of the lithium mineral reserves, may represent a setback for LIB technology. For example, around one-quarter of the global Li precursor materials are utilized to produce LIBs, thereby delivering a strong hike in the price of lithium carbonate [9,10,11]. On this premise, a wide range of research has been dedicated to alternative elements for ion battery technology [2,8,12,13,14]. Among many candidates, sodium-ion batteries (SIBs) are expected to substitute the LIBs due to their low cost, non-toxicity, and nearly limitless sodium mineral reserve [15,16]. Moreover, sodium possesses similar chemical properties as lithium since they belong to the same alkali metal group. The replacement of lithium with sodium could take advantage of the existing and mature technologies and product lines with only minimal modifications. Unfortunately, SIBs suffer from low energy storage capacity and energy density, and a low rate of charging and discharging [17].

The developmental process of ion batteries in the past few decades has advanced since the discovery of nanotechnology and nanomaterials, which provide excellently performing electrodes [18,19,20,21,22]. Nanostructures such as graphenylene, nanotubes, nanoeggs, nanocones, fullerene, and graphene have been studied for nano-electrode applications [23,24,25]. For instance, a report from Youn et al. [8] showed that graphene nanomaterials have excellent ion battery electrode properties [26]. Particularly, structural modifications such as doping with foreign materials and/or chemical functionalization of electrode nanostructures have been reported to enhance the performance of these nanomaterials [1,27,28,29]. Nnadiekwe et al. reported that boron nitride nanosheets (BNNS) have improved anodic voltage capacity when functionalized with conducting polymers such as polypyrrole [1]. Furthermore, Hardikar et al. (2014) [28] and Qie et al. [30] reported improved performance of LIB through the replacement of graphene’s carbon atoms with boron or nitrogen. Similarly, in 2013, Yu [31] reported that the performance of LIB anodic materials can be significantly improved by defective nitrogen-doping of graphene sheets. Yu established that N-substituted graphene possessing double vacancies is mostly likely to significantly enhance the performance of LIBs. Experimentally, Chen et al [32] studied the effect of N-doping of hard carbon for robust anodic material application for high-performance potassium-ion batteries (PIBs). Their report showed a high-capacity rate of 154 mAh/g at 72 cycles and a long-lasting life cycle of 4 × 10^3^ with efficient rate capability. Similar improved electrode properties, such as enhanced conductivity and voltage capacities via carbon anode N/S-codoped hierarchy for SIBs, have also been reported [33]. Commonly, less compact elements such as boron and nitrogen can yield suitable adsorption sites for ions that show weak interaction with the pristine graphene.

Graphene is a 2D nanosheet comprising sp^2^-hybridized carbon atoms. The sp^2^-hybridized bonds offer excellent structural reliability and outstanding mechanical properties, which are important for anodic material applications [34,35,36,37,38,39,40,41,42,43,44,45,46,47,48,49]. Its honeycomb extended network represents the fundamental block structure for other crucial allotropes such as stacked 3D graphite, rolled form 1D nanotubes, and the wrapped form 0D fullerenes [50]. Graphene has been extensively investigated as an anodic material [51,52,53,54] owing to its properties such as superior surface area and promising electronic properties [30]. More specifically, one-dimensional nanotubes and zero-dimensional fullerenes have been utilized as anodic materials in LIBs and have demonstrated increased electrochemical performance [51,55,56]. However, these carbon-based materials only exhibit short-lived enhanced performance when compared with 3D graphite. It is already established that the performance of nanomaterials extensively relies on their morphologies and structures. Hence, graphene nanosheets would most likely provide improved electrochemical activities. Using first-principle simulations, Gerouki et al. reported that graphene sheets of ca. 0.7 nm thick offer the best storage density with Li_4_C_6_ [57], while Hardikar et al. reported the electrochemical performance of LIBs with graphene sheets of four layers and a large specific surface area of 492.5 m^2^/g [30]. Lian et al. obtained an excellent specific capacity of about 900–1264 mAh/g for more than four layers of high-quality graphene sheets [51]. Furthermore, doping graphene support(s) with boron atom(s)/ion(s) demonstrated an improvement in the anodic performance of the support materials and, consequently, the cell voltages [58]. It has conferred on the support materials high electronic mobility, indicating potential to attain an excellent rate of performance.

Despite these efforts, there are limited reports on the intercalation of elements such as boron within the layers of high-quality graphene nanosheets for improved anodic boron-ion batteries (BIBs). Hence, in the present study, we report a theoretical investigation of the anodic performance of graphene nanosheets for boron-ion batteries. Previous reports [30,51,57] have shown that multilayer graphene with fewer layers demonstrates superior electrochemical performance for ion battery anodic applications. Therefore, we investigated the electrochemical performance of single-layer to four-layer graphene sheets. With this investigation, we aim to pave the way for the successful design of extremely effective materials for energy storage.

## 2. Theoretical Methods

All structural optimizations and electronic properties calculations were performed employing Density Functional Theory (DFT) as implemented on Gaussian 09 suite. The Perdew–Burke–Ernzerhof (PBE) functional belonging to the generalized gradient approximation (GGA) functional was used to account for the exchange–correlation energy because it provides reasonable accuracy without prolonged computational times, while the 6–311 G(d,p) basis set was adopted [8,59,60,61]. The PBE is an effective standardized functional because, by design, each component adheres to some exact conditions [62]. It follows the spin-scaling relationship exactly and reclaims the linear response of the LDA for a small-scale gradient [63]. The PBE functional is a full ab initio functional which relies on μ, β, K, and γ parametric values fixed from theoretical factors. Similarly, the 6–311 G(d,p) basis set was adopted because it provided a reliable assessment of the energies of solvation when employing an implicit solvent standard/prototype when compared to other highly revered basis sets [64]. The influence of dispersion-corrected functionals on the geometry of the layers was similarly examined using Grimme’s D3 approach [65,66,67,68]. Typically, all atoms were set free in the single- and double-layer slab. The resulting layers slab models are shown in Figure 1. However, to stabilize the computed surface energies for the three and four layers, the atoms were frozen. Studies have shown that freezing layers does not affect surface energies [59]. The adsorption energies (E_ad_) were calculated according to the following equation [69]:(1)$$E_{\mathrm{ad}}=E_{\mathrm{Complex}}-\left(E_{\mathrm{adsorbate}}+E_{\mathrm{substrate}}\right)$$
where E_complex_, E_adsorbate_, and E_substrate_ represent the energies of the absorbate and boron ion(s), graphitic layer(s), and the substrate, respectively. Likewise, the final cell voltages (V_cell_) were calculated utilizing the Nernst equation:(2)$$V_{\mathrm{cell}}=\frac{-{\Delta G}_{\mathrm{cell}}}{\mathrm{zF}}$$
where zF represents the charge on the metal ions and Faraday constant, and Δ*G*_cell_ stands for the Gibbs free energy of the overall cell reactions as
(3)$${\Delta G}_{\mathrm{cell}}={\Delta E}_{\mathrm{cell}}+P\Delta V-T\Delta S$$
where ΔG corresponds to the change in the internal energies of the cell, because the influences of entropy and the volume effects are insignificant (<0.01 V) to the cell voltage (V_cell_) [1]. We investigated the effect of the number of layers and boron ions on the cell voltage. Our model of the systems with pristine layered graphene sheets is illustrated in Figure 1. The models of the boron-intercalated layered graphene sheets are shown in Figure 2. The GaussSum program was used to depict the density of states (DOS) plot [70].

## 3. Results and Discussion

### 3.1. Structural Properties

*Pristine graphene nanosheets*: The optimized electronic structures of the graphene sheet and the different multilayer orientations of the sheets are shown in Figure 1. In the multilayer orientations, the graphene sheets were differently orientated: AB and AA for bilayer; ABA and AAB for trilayer; and ABAB, AABB, and AABA for tetralayer before optimization. However, after optimization, it was observed that the layers favored AA orientation in all the arrangements, as similarly observed by other reports [8].

*Boron-intercalated and boron-free graphene nanosheets*: Figure 2 shows the optimized structures of the boron atom and its corresponding ion (B^3+^) adsorbed on the monolayer graphene sheets (MG) (Figure 2a,b,i,j). The boron species preferred a central position on the MG after optimization even after being positioned at different points on the sheets before optimization. In the multilayered graphene orientations, the intercalated B or B^3+^ preferred the edges of the multilayer sheets after optimization. The reduced density gradient (RDG) maps for the complexes imply weak van der Waals interactions predominantly among the boron species and the graphene sheets. It may offer explanations for the preferred arrangements of the complexes. As shown in Appendix A, the TG and TTG have two boron atoms and/or ions (B^3+^) alternatively intercalated within two layers of the graphene sheets.

We examined the distances between the layers of the graphene (α-layer and β-layer) and the boron species (B/B^3+^). As shown in Figure 3, the red bars represent the distance or degree of compactness between the β-layer and the boron species. Similarly, the green bars represent the degree of compactness between the α-layer and the boron species (B/B^3+^). The monolayer graphene arrangement results from the “α-layer”-B/B^3+^ arrangement with the subsequent multilayer arrangements resulting from the successive addition of two or more graphene layers in the α- and/or β-layered direction(s). The “α-layer”-B/B^3+^ distances (the green bars) are generally shorter than the “β-layer”-B/B^3+^ distances (the red bars), probably due to stronger interaction between the α-layer and the boron species before the subsequent addition of the β-layer(s) to generate the multilayers. The β-layer distances from the boron atoms (B) are also longer than their corresponding distances from the ionic boron species (B^3+^), which further amplifies the ionic interaction effect of the boron with the graphene layer(s).

In the cases of arrangements having more than one boron atom/ion, similarly, distance trends and effects are pertinent. Generally, the presence of the boron atoms/ions intercalation influences the degree of compactness of the graphene layer arrangement; however, this may likely be insignificant to the progressive drifting of the boron atoms and/ions during the charging and discharging process of the ion battery, as illustrated in the RDG surface analysis.

### 3.2. Electronic Properties of Graphene Sheets and Absorbed B^3+^ on the Graphene Sheets

The energies of HOMO, LUMO, and HOMO–LUMO bandgap of the graphene sheets and the intercalated B^3+^ ions are calculated in Figure 4a–g and Appendix A. The presence of higher HOMO energy is a characteristic associated with donating tendencies, while low LUMO energy is considered as accepting ability.

The HOMO energies (Appendix A) of the graphitic layers relatively decrease from monolayer to tetralayer, whereas the LUMO energies recorded no significant changes. This suggests that the HOMO and LUMO levels of graphene sheets were significantly stabilized by the adsorption of the B^3+^ ion. The HOMO and LUMO levels shift from −5.085 and −2.242 eV in MG to −20.08 and −19.84 eV in 2B^3+^@TTG. Although there are no significant differences between the energies of the HOMO and LUMO of the singly adsorbed B^3+^ ions, relative variations are seen with doubly adsorbed B^3+^ ions: −20.08 and −19.84 eV for 2B^3+^@TG and −18.33 and −17.66 eV, corresponding to 2B^3+^@TTG, which can be attributed to the addition of B^3+^ into the graphene sheets. Accordingly, upon adsorption of B^3+^, the global electronic energy bandgap (E_g_) of the graphene sheets is reduced significantly to about 99.89%, and the same trend is observed with adsorbed neutral boron (Appendix A).

The observed order with adsorbed neutral boron could be due to the existence of unpaired electrons in the valence shell of the neutral boron; the HOMO level of the graphitic layer is largely impacted by altering to higher energies, suggesting a considerable destabilization [71]. Similarly, the E_g_ of the doubly adsorbed B^3+^ increases in comparison with the singly adsorbed B^3+^, which could be attributed to the decreased LUMO level and increased HOMO level. In electrostatic potential maps (ESPs), colors are used as indicators for different electrostatic potential values: blue demonstrates high positive (electron-deficient) regions of the species, while green displays the region of zero potential [72]. As illustrated in Appendix A, the ESPs of the B^3+^ form of B^3+^@MG, B^3+^@BG, B^3+^@TG, B^3+^@TTG_asym, B^3+^@TTG_sym, 2B^3+^@TG, and 2B^3+^@TTG reveal predominantly more electropositive regions (blue color). This is more severe with B^3+^@MG, B^3+^@BG, 2B^3+^@TG, and 2B^3+^@TTG configurations. However, with the addition of layers, a slightly neutral region with B^3+^@TG, B^3+^@TTG_asym, and B^3+^@TTG_sym was observed.

### 3.3. Adsorption of B^3+^ Ions on the Wall of Graphene Sheets

The adsorption of B^3+^ ions on the wall of the solid host is considered an alternative B^3+^-storage mechanism. The model sheets studied with adsorbed B^3+^ ions are depicted in Figure 2. The role of the numbers of graphene sheets and the number of B^3+^ ions are examined. We studied the adsorption characteristics of graphitic carbon sheets by intercalating B^3+^ ions between the bilayer, trilayer, and tetralayer (Figure 2). The positions of the intercalated B^3+^ ions between the graphene sheets were selected thoroughly. All the B^3+^ ions were placed in the central plane defined by the graphene sheet and allowed to relax in all directions. 

The adsorption energies (E_ad_) for the singly intercalated B^3+^ ions were calculated as 3.987, −320.1, −477.0, −638.4, and −637.6 kcal/mol for B^3+^@MG, B^3+^@BG, B^3+^@TG, 2B^3+^@TTG_asym, and B^3+^@TTG_sym, respectively (Figure 5). Furthermore, the tetralayer graphitic carbon B^3+^@TTG_asym and B^3+^@TTG_sym exhibited the most favorable and stable adsorption configuration of B^3+^ intercalated within different layers of graphene sheets. Similarly, the E_ad_ corresponding to the formation of 2B^3+^@TG and 2B^3+^@TTG configurations having double B^3+^ ion intercalations are −467.9 and −628.7 kcal/mol, respectively. Thus, the addition of an extra sheet to form 2B^3+^@TTG was more energetically favored than the formation of the 2B^3+^@TG with fewer layers. Meanwhile, a very weak interaction was observed on B^3+^@MG with an E_ad_ of 3.987 kcal/mol, in agreement with some research findings on Li [73]. 

The reduced density gradient (RDG) isosurfaces of the interactions are shown in Figure 6 and Appendix A. Bader’s theory of atoms in molecules categorized van der Waals interactions, strong steric effects, and hydrogen bond interactions to exhibit low *ρ* and relatively larger *ρ* values, respectively [74]. RDG isosurfaces plots reveal the binding interaction region and the modes [75]. The RDG isosurfaces analysis results of BC_6_ and BC_2_ mode of binding interactions manifest the weak van der Waals interaction between B^3+^ and graphene sheets, as displayed in Figure 6 and Appendix A, suggesting the physical nature of the reaction and high diffusion affinity of the B^3+^ ions [76]. Except for B^3+^@MG, in which the intercalated B^3+^ is equidistant with six carbons of the graphene ring to form BC_6_ (Figure 6a,b), all other configurations revealed the BC_2_ geometry, where the B^3+^ ions are adjacent to two carbon atoms, one above and one below the graphene layer (Figure 6c–f and Appendix A).

To gain in-depth knowledge of the interactions between the intercalated B^3+^/B atom and the numbers of graphene sheets, we calculated the density of states (DOS) and the projected density of states (PDOS) of the B^3+^/B atom adsorbed graphene sheets as displayed in Figure 7 and Appendix A. The projected density of states (PDOS) plots revealed the reduction in E_g_ from 2.843 eV in MG to 2.472 eV with TTG, which is likely due to an increase in the number of graphene sheets (Figure 7). Similarly, as seen in the PDOS and the frontier molecular orbital, the number of states at the HOMO is influenced by the carbon atoms, while the contribution at the LUMO increases with an increase in the graphene sheet. In general, the valence and conduction bands of MG, BG, TG, and TTG are dominated by the p orbitals of both types of carbon atoms (either bonded with C or H). Unlike the boron atoms, the adsorption of B^3+^ significantly reduces the levels of HOMO and LUMO in the graphene sheets and limits their E_g_, which would simplify the diffusion of electrons or charges. Analysis of the PDOS and frontier molecular orbital implies that the LUMO is localized on the B^3+^ while the HOMO is confined on the graphitic sheets but improved with the addition of the sheets. Furthermore, the adsorption of two B^3+^ into TG and TTG followed the same trend. However, upon adsorption of boron, a new electronic peak (indicated by the red arrow) is generated, which mainly comes from the contribution of 2p unpaired electrons of the B atom (Appendix A). The presence of unpaired electrons at the HOMO generates unstable singly occupied molecular orbitals, resulting in the lowering of the HOMO from −5.085 eV in B@MG to 0.040 eV in 2B@TG (Appendix A). In general, the PDOS and the frontier molecular orbital analysis illustrate that the contribution of the number of states at the HOMO is mainly influenced by the boron atoms, while the LUMO is controlled by the atoms of the graphene sheet.

Natural bond orbitals (NBO) population analysis was conducted to estimate the charge transfer in the complexes and is presented in Appendix A. As shown in Table 1, the complexes exhibit σ donation between the graphene sheets and the B^3+^ ion, and π back-donation from the B^3+^ ion to the vacant orbitals of the graphene atoms. The NBO results likewise indicate that the charge transfer is more sensitive to the sum of adsorbed B^3+^ ions in 2B^3+^@TG and 2B^3+^@TTG than the other complexes, suggesting greater stability in the B^3+^@MG, B^3+^@BG, B^3+^@TG, B^3+^@TTG_asym, and B^3+^@TTG_sym configurations in contrast to the 2B^3+^@TG and 2B^3+^@TTG complexes.

### 3.4. Ion Battery Applications

The graphene sheets (single and multilayers) were considered as anodic electrodes for the boron-ion batteries with the half-cell reactions taking place at the cathode and anode shown in Equations (4)–(7). Equation (8) also shows the overall cell reaction.
Cathode: B^3+^ + 3e^−^ → B(4)
Anode: B@sheet → B^3+^@sheet + 3e^−^(5)
Charging Process: B^3+^@sheet + 3e^−^ → B@sheet(6)
Discharging Process: B@sheet → B^3+^@sheet + 3e^−^(7)
Overall cell reaction: B^3+^ + B@sheet → B^3+^@sheet + B + ΔG_cell_(8)
where B/B^3+^ is the boron atom/ion and “sheet” is the monolayer graphene sheet (MG), bilayer graphene sheet (BG), trilayer graphene sheet (TG), and tetralayer graphene sheet (TTG). The cell voltage (V_cell_) and the cell energy are computed according to Equations (2) and (3).

As depicted in Figure 5, the adsorption energies of the boron atom(s) and/or ion(s) with the graphene sheets increased negatively as the number of the layers increased, with the boron ion (B^3+^) adsorption being more negative than the atomic counterparts. The monolayer graphene sheet (B^3+^/B@MG) has the least negative adsorption energy, while the tetralayer graphene sheet (B^3+^/B@TTG) showed the highest negative adsorption energy. The complexes with more negative adsorption energies express more favorable and deeper van der Waals interactions [65,77]. Table 2 shows the overall cell energies (ΔE_cell_) and the corresponding voltages of the cells. The cells’ energy increases with an increasing number of graphene sheets. However, for the intercalation of two atoms/ions of boron, the overall cell adsorption energy is higher. This can be attributed to the increased weak interactions between the graphene sheets and the two boron ions, which still permit free movement of the boron ions within the graphene layers during the charging and discharging processes.

Moreover, compared to sodium and lithium-ion batteries (SIBs and LIBs) [1] and other battery materials reported (Table 3), the voltages obtained here are significantly improved. The B^3+^/B@TTG showed the highest cell voltage of 16.5 V, which indicates that in addition to the complexes being efficient alternatives for the anodic electrode of boron-ion batteries, increasing the number of the graphene layers also improved the electrochemical performance of the storage system. Generally, the boron-graphene sheets have shown excellent voltage outputs, hinting at their suitability as anodic material(s) for enhancing the energy storage performance of boron-ion batteries (BIBs) if extensively studied.

## 4. Conclusions

We have studied the high-energy-density anodic compartment for boron-ion batteries as an alternative to lithium-ion and sodium-ion batteries using the first-principles calculations within the framework of Density Functional Theory. We investigated the electrochemical performance of the boron ion(s) on the monolayer, bilayer, trilayer, and tetralayer graphene sheet electrodes. Significant decreases in the HOMO–LUMO energy gap from −5.085 eV to −2.242 eV for B^3+^@MG and from −20.08 eV to −19.84 eV for 2B^3+^@TTG were recorded. The predominant interaction force for the graphene layers was van der Waals, as predicted by the reduced density gradient isosurface analyses, with the B^3+^@TTG_asym and B^3+^@TTG_sym configurations showing the most favorable interactions. 

Furthermore, the electrochemical cell voltages obtained with the single-layer (B^3+^/B@MG) and/or multilayer (B^3+^/B@BG, B^3+^/B@TG, and B^3+^/B@TTG) graphene sheets were significantly improved, with B^3+^/B@MG (13.7 V) showing the least voltage. On the contrary, B^3+^/B@TTG (16.5 V) showed the highest voltage. The results revealed that increasing the number of graphene layers improves the electrochemical performance of the anodic electrode of the boron-ion battery. Therefore, the number of layers or the thickness of the graphene nanosheets is another effective parameter to tune the anode performance for boron-ion batteries. This might suggest an additional dimension to further engineer the graphene anode performance by increasing the number of graphene layers. Our theoretical investigations demonstrate the suitability of the graphene-based anodic electrodes for boron-ion batteries when extensively investigated for large-scale application as a highly efficient energy storage substitute for lithium- and sodium-based ion batteries, which could be beneficial to the material design of ion batteries.

## Figures and Tables

**Figure 1 nanomaterials-12-01280-f001:**
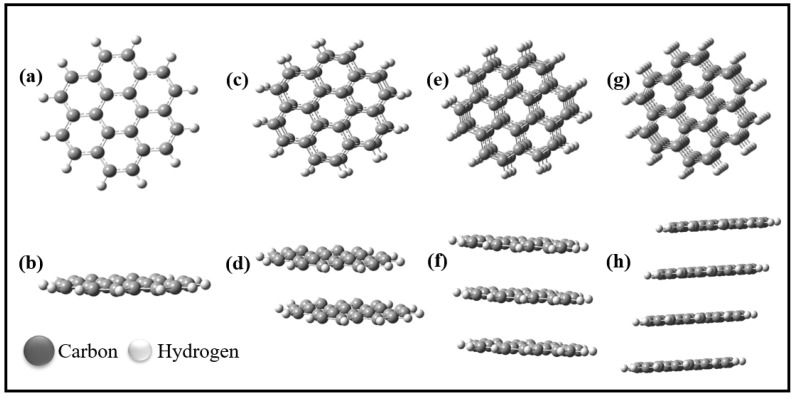
Optimized structures of (**a**,**b**) monolayer graphene sheet (MG), (**c**,**d**) bilayer graphene sheet (BG), (**e**,**f**) trilayer graphene sheet (TG), and (**g**,**h**) tetralayer graphene sheet (TTG). (**b**,**d**,**f**,**h**) Side-view orientation of the graphene sheets.

**Figure 2 nanomaterials-12-01280-f002:**
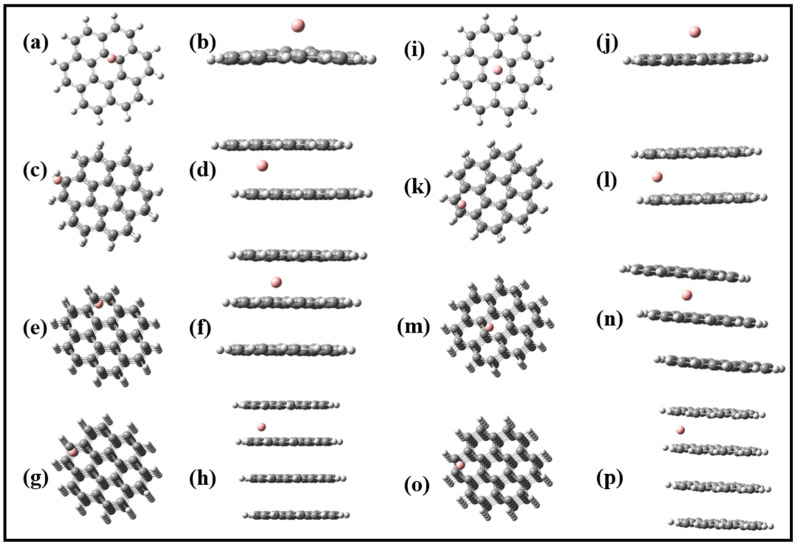
Optimized structures of (**a**,**b**) B@MG (side and top views), (**c**,**d**) B@BG (side and top views), (**e**,**f**) B@TG (side and top views), (**g**,**h**) B@TTG (side and top views), (**i**,**j**) B^3+^@MG (side and top views), (**k**,**l**) B^3+^@BG (side and top views), (**m**,**n**) B^3+^@TG (side and top views), and (**o**,**p**) B^3+^@TTG (side and top views).

**Figure 3 nanomaterials-12-01280-f003:**
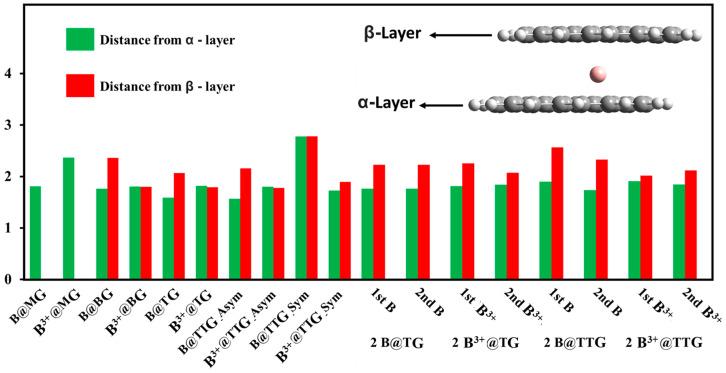
The distance between the graphene layers (α-layer and β-layer) and the boron species (B and/or B^3+^). The green bars represent the “α-layer”-B/B^3+^ distance while the red bars represent the “β-layer”-B/B^3+^ distance.

**Figure 4 nanomaterials-12-01280-f004:**
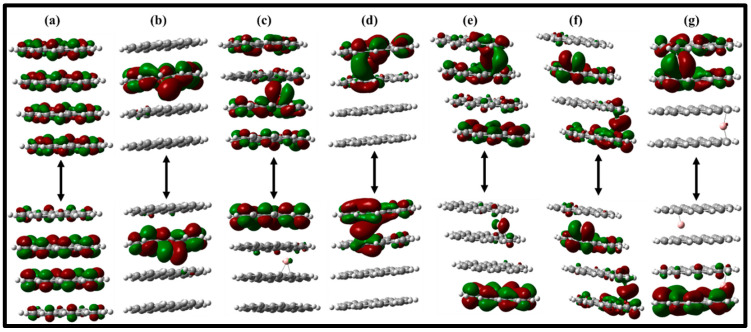
Molecular orbital HOMO (Lower) and LUMO (upper) of (**a**) tetralayer graphene sheet (TTG), (**b**) B@TTG-sym, (**c**) B^3+^@TTG-sym, (**d**) B@TTG-asym, (**e**) B^3+^@TTG-asym, (**f**) 2B@TG (side and top views), and (**g**) 2 B^3+^@TTG (side and top views).

**Figure 5 nanomaterials-12-01280-f005:**
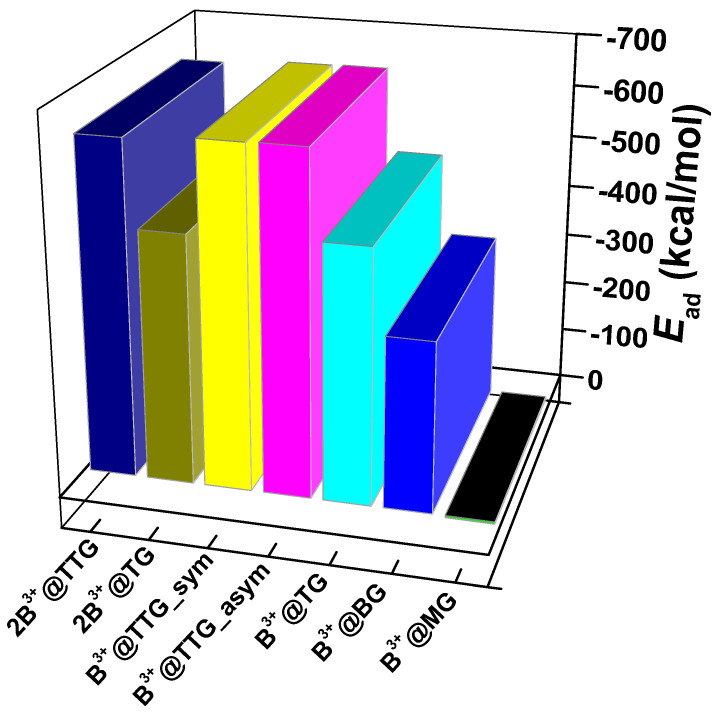
Adsorption energies (E_ad_) of B^3+^ ions on the seven studied systems including B^3+^@MG, B^3+^@BG, B^3+^@TG, B^3+^@TTG–asym, B^3+^@TTG–sym, 2B@TG, and 2 B^3+^@TTG.

**Figure 6 nanomaterials-12-01280-f006:**
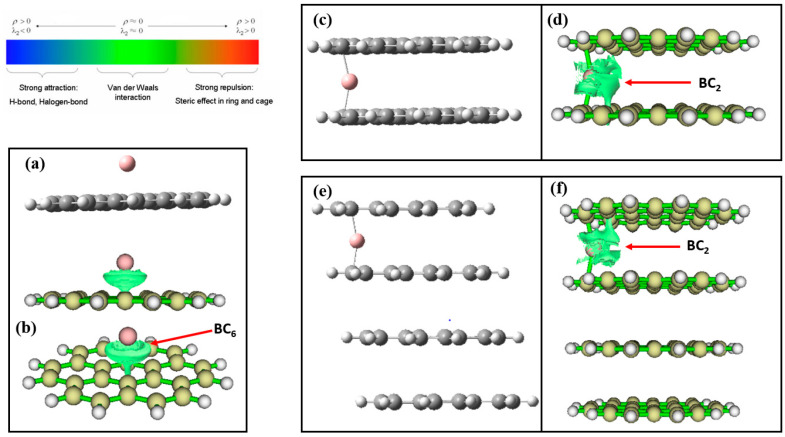
The reduced density gradient (RDG) isosurfaces analyses of (**a**,**b**) B^3+^@MG (complex and RDG), (**c**,**d**) B^3+^@BG (complex and RDG), (**e**,**f**) B^3+^@TTG (complex and RDG). Blue regions correspond to strong hydrogen bonds; red regions indicate strong steric effects; green regions describe strong van der Waals interactions.

**Figure 7 nanomaterials-12-01280-f007:**
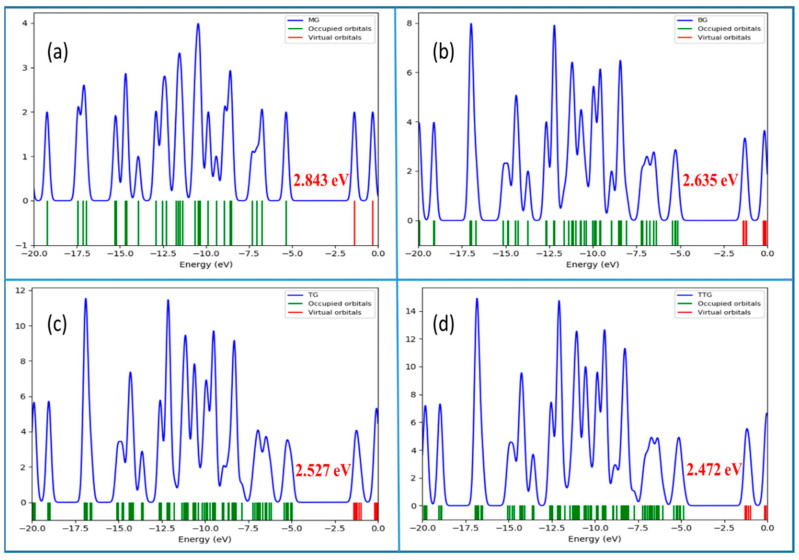
Partial density of states (PDOS) plots of (**a**) MG (**b**) BG, (**c**) TG, and (**d**) TTG.

**Table 1 nanomaterials-12-01280-t001:** Summary of NBO analysis of π-complexation between B^3+^ ions and the graphene sheets.

Complexes	C → B^3+^ Interaction (σ Donation)	B^3+^ → C Interaction (p−π* Back-Donation)	Total Charge
B^3+^@MG	0.645	−0.049	0.596
B^3+^@BG	0.971	C1 = −0.413 C2 = −0.409	0.560
B^3+^@TG	1.082	C1 = −0.251 C2 = −0.242	0.836
B^3+^@TTG_asym	0.916	C1 = −0.443 C2 = −0.429	0.480
B^3+^@TTG_sym	0.981	C1 = −0.491 C2 = −0.219	0.626
2B^3+^@TG	B^3+^1 = 1.212 B^3+^2 = 0.994	C1 = −0.328 C2 = −0.210 C1 = −0.323 C2 = −0.237	0.630
2B^3+^@TTG	B^3+^1 = 1.002 B^3+^2 = 1.017	C1 = −0.443 C2 = −0.400 C1 = −0.325 C2 = −0.341	0.633

C1 and C2 represent the shortest distant carbon of the graphene sheet above or below, respectively, and B^3+^ is the adsorbed boron ion, π* represent the anti-bonding π-orbitals.

**Table 2 nanomaterials-12-01280-t002:** The overall cell energy change (ΔE_cell_) and the cell voltages of the graphene sheet(s).

	ΔE_cell_ (kcal/mol)	V_cell_ (Volts)
B^3+^/B@MG	1.513	13.7
B^3+^/B@BG	1.701	15.4
B^3+^/B@TG	1.742	15.8
B^3+^/B@TTG_asym	1.815	16.5
B^3+^/B@TTG_sym	1.786	16.2
2B^3+^/2B@TG	2.723	12.4
2B^3+^/2B@TTG	2.959	13.4

**Table 3 nanomaterials-12-01280-t003:** Comparison of the present results with those reported in the literature.

S/N	Support(s)	Dopant(s)	Cell Voltage(s)	Ref.
1	Boron nitride sheet	Li/Li^+^, Be/Be^2+^, Na/Na^+^, Mg/Mg^2+^, and polypyrrole	Li-ion = 2.06 V Na-ion = 1.37 V Be-ion = 6.60 V Mg-ion = 3.82 V	[1]
2	Boron nanorod	Li/Li^+^	Li-ion = 2.00 V	[78]
3	Magnesium polymorphs	Na/Na^+^	Na-ion = 1.5 V	[79]
4	Inorganic boron nitride nanocluster	Na/Na^+^, F^−^, Cl^−^, and Br^−^	Na-ion = 3.39 V	[80]
5	Boron nitride nanosheet	Na/Na^+^, P, and Al	Na-ion = 2.31 V	[12]
6	Hexagonal boron phosphide	Li/Li^+^, Na/Na^+^, K/K^+^	Li-ion = 1.37 V Na-ion = 0.97 V K-ion = 0.93 V	[81]
7	Aluminum/boron phosphide nanocluster	Na/Na^+^, F^−^, Cl^−^, and Br^−^	Na/F@B_12_P_12_ = 4.5 V Na/Cl@B_12_P_12_ = 3.47 V Na/Br@B_12_P_12_ = 3.5 V Na/F@Al_12_P_12_ = 3.5 V Na/Cl@Al_12_P_12_ = 3.22 V Na/Br@Al_12_P_12_ = 3.20 V	[82]
8	Phosphorene	Mg/Mg^2+^	Mg-ion = 0.833 V	[83]
9	Graphene-like MoS_2_ cathode and ultrasmall Mg nanoparticles	Mg/Mg^2+^	Mg-ion = 1.8 V	[84]
10	B_40_ fullerenes	Mg/Mg^2+^, F^−^, Cl^−^, and Br^−^	Mg-ion = 8.8 V	[85]
11	Multilayer graphene sheet	B/B^3+^	B-ion = 16.5 V	Present study

## Data Availability

The raw/processed data required to reproduce these findings cannot be shared at this time due to time limitations and will be made available by the corresponding author upon request.

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
