# Peer review of "A First-Principles Study on the Multilayer Graphene Nanosheets Anode Performance for Boron-Ion Battery"

_nanomaterials, 2022, doi:10.3390/nano12081280_

Round 1
Reviewer 1 Report
In the Manuscript authors by Umar et al ., the authors reported a computational study of 2D anode materials used for Li-ion batteries. Specifically, the authors focused on Boron-doped graphene. Their density functional theory calculations predicted the electrochemical performance of the boron insertion by predicting properties such as HOMO-LUMO gap, cell voltage, etc. Based on the calculations, the authors concluded that increasing the number of graphene layers improves the electrochemical performance of the anodic electrode of the boron-ion battery. I have two major comments for the authors to consider.
- I suggest the authors add more discussions of the importance of the realizability of boron insertion into graphene
- I suggest the authors include more discussion about the benchmark of DFT functionals.
Author Response
Reviewer 1
In the Manuscript authors by Umar et al., the authors reported a computational study of 2D anode materials used for Li-ion batteries. Specifically, the authors focused on Boron-doped graphene. Their density functional theory calculations predicted the electrochemical performance of the boron insertion by predicting properties such as HOMO-LUMO gap, cell voltage, etc. Based on the calculations, the authors concluded that increasing the number of graphene layers improves the electrochemical performance of the anodic electrode of the boron-ion battery. I have two major comments for the authors to consider.
I suggest the authors add more discussions of the importance of the realizability of boron
insertion into graphene.
Response: We thank the reviewer for the suggestion. We have incorporated the importance of the realizability of boron insertion into graphene in the revised manuscript as advised. See page 3, line 97-101).
I suggest the authors include more discussion about the benchmark of DFT functionals.
Response: The authors appreciate the reviewer. Please kindly note that the detailed discussions on the benchmark of DFT functionals is out of the scope of our study. However, we have included further discussions on the justification for the choice of the PBE functional have been included in the revised manuscript on page 4, lines 117-124.

Reviewer 2 Report
The manuscript was written by Mustapha Umar entitled “A First-principles Study on the Multilayer Graphene Nanosheets Anode Performance for Boron-ion Battery” can be published in the journal of Nanomaterials subject to minor revision.
- The simulated interaction (adsorption) energies are very large; is it possible for the author to simulate the Counterpoise corrected energies of these systems?
- The authors can help from these papers
a. Systematic Analysis of Poly(o‑aminophenol) Humidity Sensors. ACS Omega, 2017, 6380-6390.
b. Journal of Molecular Structure Volume 1127, 5 January 2017, Pages 734-741
- In order to have a deeper insight into the B3+@TTG I will also recommend simulating the NBO and electrostatic potential map (ESP) of the mentioned species. Again, authors can help from the provided literature.
- Besides, there are some minor typo mistakes and they should be corrected before resubmission.
Author Response
Reviewer 2
The manuscript was written by Mustapha Umar entitled “A First-principles Study on the Multilayer Graphene Nanosheets Anode Performance for Boron-ion Battery” can be published in the journal of Nanomaterials subject to minor revision.
- The simulated interaction (adsorption) energies are very large; is it possible for the author to
simulate the Counterpoise corrected energies of these systems?
Response: We thank the reviewer for the observation. We agree with the reviewer that the simulated interaction energies are large. This could be attributed to the strong Van der Waals attraction between the sheets and the boron ions. Also note that dispersion correction which takes care of the long-range interactions was implemented using the Grimme’s DFT-D3 for calculating the energies. Having said that, we attempted to run the counterpoise but the timeframe for the revision isn’t enough to achieve this as the systems are quite bulky to converge within this period. We will take this into considerations in future reports.
- The authors can help from these papers
- Systematic Analysis of Poly(o-aminophenol) Humidity Sensors. ACS Omega, 2017, 6380-6390.
b. Journal of Molecular Structure Volume 1127, 5 January 2017, Pages 734-741.
Response: We thank the reviewer for the suggestions.
- In order to have a deeper insight into the B3+@TTG I will also recommend simulating the NBO
and electrostatic potential map (ESP) of the mentioned species. Again, authors can help from the
provided literature.
Response: We have conducted the NBO analysis, and also included the ESP maps of the systems. These are included in the revised manuscript on page 12-13, lines 310-327, and new supplementary figures S8 and S9.
- Besides, there are some minor typo mistakes and they should be corrected before resubmission.
Response: The manuscript has been thoroughly revised and the typos corrected accordingly.

Reviewer 3 Report
In this paper, the properties of B3+ on single-layer (mg), double-layer (BG), three-layer (TG) and four layer (tTG) graphene sheets were studied by first principle calculation. The cell voltage obtained were considerably enhanced and B 3+ /B@TTG showed the highest cell voltage of 16.5 V. Our results suggest a novel avenue to engineering the graphene anode performance by increasing the number of the graphene layers. This paper is well written, but needs the following revisions before publication:
- What are the advantages of this job over other jobs? The author is advised to make a table for comparison.
- There are too many pictures in the text. The author needs to put the pictures that are not particularly important in the attachment
- There is something wrong with the number of the formula. The author should express it in another form.
- The information of many figures in the article is not clear, and the author needs to redraw them.
- About “Graphene”, some relevant literature authors need to mention, such as: RSC Adv., 2022, 12, 7821-7829; Talanta, 134(2015) 435–442; RSC Adv., 8(2018) 42233–42245; ACS Sustain. Chem. Eng., 3(2015) 1677–1685; RSC Adv., 7(2017) 25314–25324.
Author Response
Reviewer 3
In this paper, the properties of B3+ on single-layer (mg), double-layer (BG), three-layer (TG) and four layer (tTG) graphene sheets were studied by first principle calculation. The cell voltage obtained were considerably enhanced and B 3+ /B@TTG showed the highest cell voltage of 16.5V. Our results suggest a novel avenue to engineering the graphene anode performance by increasing the number of the graphene layers. This paper is well written, but needs the following revisions before publication
- What are the advantages of this job over other jobs? The author is advised to make a table for
comparison.
Response: We thank the reviewer for raising such important point. We have included a new Table (Table 3) which shows the merits of the present study by comparison with other reported studies.
- There are too many pictures in the text. The author needs to put the pictures that are not
particularly important in the attachment
Response: The figures have been revised and some of them have been moved to the supplementary file.
- There is something wrong with the number of the formula. The author should express it in another form.
Response: These have been corrected in the revised version.
- The information of many figures in the article is not clear, and the author needs to redraw them.
Response: These have been taken care of in the revised version.
- About “Graphene”, some relevant literature authors need to mention, such as: RSC Adv., 2022,
12, 7821-7829; Talanta, 134(2015) 435–442; RSC Adv., 8(2018) 42233–42245; ACS Sustain.
Chem. Eng., 3(2015) 1677–1685; RSC Adv., 7(2017) 25314–25324.
Response: The relevant references have been incorporated into the manuscript on page 3 line 80.

Round 2
Reviewer 3 Report
The article has been modified by the system and can be received.